# Facility Managers’ Perceptions of Support and Supervision of Ward Based Primary Health Care Outreach Teams in National Health Insurance Pilot Districts in KwaZulu-Natal, South Africa. A Qualitative Study

**DOI:** 10.3390/healthcare9121718

**Published:** 2021-12-13

**Authors:** Euphemia Mbali Mhlongo, Elizabeth Lutge

**Affiliations:** 1Department of Nursing, School of Nursing and Public Health, Howard College, University of KwaZulu Natal, Durban 4000, South Africa; elizabeth.lutge@kznhealth.gov.za; 2KwaZulu-Natal Department of Health, Pietermaritzburg 3200, South Africa

**Keywords:** supervision, community health workers, community caregivers, ward-based outreach teams, national health insurance, challenges, successes

## Abstract

Introduction: Evidence from many countries suggests that provision of home and community-based health services, linked to care at fixed primary health care facilities, is critical to good health outcomes. In South Africa, the Ward-Based Primary Health Care Outreach Teams are well placed to provide these services. The teams report to a primary health care facility through their outreach team leader. The facility manager/operational manager provides guidance and support to the outreach team leader. Aim: The aim of the study was to explore and describe the perceptions of facility managers regarding support and supervision of ward-based outreach teams in the National Health Insurance pilot sites in Kwa Zulu-Natal. Setting: The study was carried out in three National Health Insurance pilot districts in KwaZulu- Natal. Methods: An exploratory qualitative design was used to interview 12 primary health care facility managers at a sub-district (municipal) level. The researchers conducted thematic analysis of data. Findings: Some gaps in the supervisory and managerial relationships between ward based primary health care outreach teams and primary health care facility managers were identified. High workload at clinics may undermine the capacity of PHC facility managers to support and supervise the teams. Field supervision seems to take place only rarely and for those teams living far away from the clinic, communication with the clinic manager may be difficult. The study further highlights issues around the training and preparation of the teams. Conclusions: Ward based primary health care outreach teams have a positive impact in preventive and promotive health in rural communities. Furthermore, these teams have also made impact in improving facility indicators. However, their work does not happen without challenges.

## 1. Introduction

South Africa (SA) is experiencing a quadruple disease burden made up of HIV/AIDS and TB, maternal and childhood diseases, non-communicable diseases and violence and injuries [1]. The high disease burden is compounded by poverty related factors such as lack of access to health care facilities. Human resource for health in South Africa are limited and inequitably distributed, and this undermines the efforts made to reduce the burden of disease [1,2]. Although South Africa is considered a middle-income country, the country’s health outcomes are poor and there is an urgent need for a health care system that will strike the balance between preventive, promotive, curative and rehabilitative services with suitable referral systems in place. There is a need for a category of health workers who will meet the basic health needs of people at household level to ensure accessible, affordable health care using appropriate technologies acceptable to the recipients of care [3]. At the beginning of democracy (in 1994 South Africa received first democratic elections), primary health care was chosen as the primary mode of healthcare delivery in South Africa, focusing on prevention of disease and promotion of health. In addition, it is one of the pillars of the National Health Insurance (NHI), implemented in South Africa since 2014. The ward based primary health care outreach teams (WBPHCOTs) are well placed to provide community based primary health care services which encompass activities in communities, households and supported by referral networks to the clinics and hospitals. Community health workers had their origins in China in the 1920s and they were precursors of the “barefoot doctor” program, a movement in the 1950s. In the 1960s community health worker programs emerged in Indonesia, India, Tanzania and Venezuela [3,4]. Community health worker programs proved to be successful in the 1980s in countries such as Brazil, Bangladesh, and Nepal [3].

In 2010, South Africa’s National Department of Health (NDoH) launched a national primary health care (PHC) initiative to strengthen health promotion, disease prevention, and early disease detection. The strategy, called re-engineering Primary Health Care (rPHC), aims to support a preventive and health-promoting community-based PHC model by using community-based outreach teams (known in South Africa as Ward-based Primary Health Care Outreach teams). These teams provide health education, promote healthy behaviors, assess community health needs, manage minor health problems, and support linkages to health services and health facilities [4]. The WBPHCOTs are staffed by Community health workers (CHWs) and they form a fundamental part of rPHC in South Africa [5]. The ward based primary health care outreach teams (WBPHCOTs) are well placed to provide community based primary health care services which encompass activities in communities, households and supported by referral networks to clinics and hospitals. These WBPHCOTs are staffed by Community health workers (CHWs)/community caregivers (CCGs) under the supervision of the outreach team leader who is a facility based professional nurse, in some instances an enrolled nurse. Research studies advise that effective supervision of CHWs can motivate them, and build a sense of legitimacy for both CHWs and the community they serve.

Studies further suggest that good supervision can identify and correct poor CHW practices, and help resolve the challenges that CHWs encounter [6]. Supportive supervision is crucial in CHW programmes, where team leaders need to play roles of facilitating, mentoring and coaching. The outcome of supportive supervision can be improved motivation and performance [7]. Regardless of the recognized role that supportive supervision can play in performance and motivation, studies have found that supervision often has low coverage, low administrative focus, is irregular, unsupportive, and demotivating, with inadequate training for supervisors and problem solving or feedback mechanisms for providers [6]. Facility managers and team leaders are well placed to train, empower and provide day to day support to CHWs.

This article is part of a larger study conducted in the three National Health Insurance (NHI) pilot districts in KwaZulu-Natal. The study participants were primary health care supervisors, facility managers, team leaders, community health workers and household members. This paper seeks to describe the facility managers’ perceptions of supervising the WBPHCOTs in the three National Health Insurance pilot districts in KwaZulu-Natal.

### 1.1. Aim of the Study

The aim of the study was to explore and describe the perceptions of facility managers regarding support and supervision of ward-based outreach teams in the National Health Insurance pilot sites in Kwa Zulu-Natal.

### 1.2. Definition of Key Concepts

#### 1.2.1. Supportive Supervision

In this article, supportive supervision is operationally defined as supervision CHWs receive from facility managers and their team leaders in the WBPHCOTs programme. Community health workers appreciate guidance and support they receive from the facility managers.

#### 1.2.2. National Health Insurance

“National Health Insurance (NHI) is a health care financing system that is designed to pool funds to actively purchase and provide access to quality, affordable personal healthcare services for all South Africans based on their health needs, irrespective of their socioeconomic status. NHI is intended to move South Africa towards Universal Health Coverage (UHC) by ensuring that the population has access to quality health services and that it does not result in financial hardships for individuals and their families”.[8]

#### 1.2.3. Municipal Ward Based Primary Health Care Outreach Teams (WBPHCOTs)

The outreach teams consist of Community Health Workers (CHWs), led by a nurse, and are linked to a PHC facility. CHWs assess the health status of individuals in the households. They provide health promotion education, identify those in need of preventive, curative or rehabilitative services, and health related counselling, and refer those in need of services to the relevant PHC facility [8].

#### 1.2.4. Operation Sukuma Sakhe

Operation Sukuma Sakhe (OSS) is a provincial programme that was founded on the premise of taking government to the people in a coordinated manner. It is a Zulu phrase which means “stand up and build”. It involves co-ordination amongst sector departments and prioritises households that need urgent interventions [9].

#### 1.2.5. War Rooms

“War Rooms are defined as integrated service delivery structure comprised of government, municipality, Community Based Organisations, private businesses and other stakeholders at ward level. Operation Sukuma Sakhe war rooms are comprised of officials from various sector departments. The different regions within the department must ensure that they have a representative attending war rooms respectively”.[9]

## 2. Material and Methods

### 2.1. Research Design

An exploratory qualitative research design was used to explore the facility managers’ perceptions of supporting and supervising ward based primary health care outreach teams. This research formed part of a larger study entitled Stakeholders’ Perceptions on the effectiveness of Ward Based Primary Health Care Outreach Teams in the Three National Health Insurance Pilot Districts in Kwa Zulu-Natal.

### 2.2. Research Site

The study was conducted in three National Health Insurance (NHI) pilot districts in KwaZulu-Natal namely, Umzinyathi, Umgungundlovu and Amajuba districts. These three districts have been piloting the National Health Insurance since 2014 in KwaZulu-Natal province. This research forms part of a larger study on the Stakeholders’ Perceptions on the effectiveness of Ward Based Primary Health Care Outreach Teams in the three National Health Insurance Pilot Districts in Kwa Zulu-Natal All three districts are predominantly rural in nature, although UMgungundlovu does include a large urban area. The health facilities’ challenges consist of shortage of staff, resources and poor infrastructure [10]. There is limited access to primary health care facilities caused by long distances, transport unavailability, poor geographic terrain and financial constraints [10]. This is common in most of the rural districts in South Africa.

### 2.3. Selection of Participants and Data Collection

The research participants were the facility managers at primary health care facilities who supervised ward based primary health care outreach teams for 3 months or longer. Facility managers who managed and supervised WBPHCOTs for three months or longer were purposively selected in each district, in order to ensure that respondents had a minimum level of experience in this role. Individual interviews were conducted with 12 facility managers (see Table 1 below) using a semi-structured interview tool (Interview schedule added as a Appendix A). Data collection took place from July 2018 to February 2019 across the three National Health insurance pilot districts in KwaZulu-Natal. The interviewers included the first author and a research assistant who was studying for a doctoral degree in public health. One week training on the study and data collection methods was conducted for the research assistant prior to commencing data collection. A comprehensive set of 19 open ended questions and relevant prompts was developed. These addressed four key thematic areas related to support and supervision but maintained sufficient flexibility to allow for emerging themes to be evoked. Interviews took place in a private room in the PHC clinics and were conducted in IsiZulu. The duration for each interview was 45 min to an hour. All participants received an information sheet explaining the purpose of the study. The study objectives were explained and participants were assured of confidentiality. Participants were given time to ask questions and thereafter provided written informed consent. Consent for digital audio recording of interviews was also obtained by signing the necessary consent forms. All interviews were recorded. The interviews continued until data saturation was reached [11]. We used the following to identify data saturation: “New data tend to be redundant of data already collected. In interviews, when the researcher begins to hear the same comments again and again, data saturation is being reached. It is then time to stop collecting information and to start analysing what has been collected” [11].

### 2.4. Data Analysis

The interviews were transcribed verbatim using Microsoft Word. Verbatim transcription with non-verbal cues of behaviour are necessary to establish reliability, dependability and trustworthiness of the study [12,13]. The transcripts were transcribed to English by an independent researcher fluent in both IsiZulu trustworthiness and English languages. Same were back translated to IsiZulu to confirm accuracy of the original translation. The study used steps in qualitative data analysis as defined by Creswell (2018:193–198): in order to organize and prepare data for analysis, read through all the data, start coding all of the data, generate a description and themes, representing the description and themes and interpreting the meaning of themes/descriptions [14]. The research team listened to the audio tapes, transcribed data to generate transcripts. The first author and the research assistant immersed themselves in the data by reading and re-reading transcripts. Categories and themes were manually assigned to the data.

### 2.5. Trustworthiness

Trustworthiness in qualitative studies is when the findings of the study accurately represent the experiences of the population being studied [15]. Measures to ensure trustworthiness as highlighted by Lincoln and Guba (1985) formed the present study. [4] Credibility was achieved through prolonged engagement with the facility managers during data collection. The use of focus groups and individual interviews also increased the credibility of the study, as the different views and experiences of participants were confirmed against others [16]. Data were collected from different districts and sub-districts (site triangulation), in KwaZulu-Natal. As similar results emerged from these different sites, credibility was enhanced. Participants were informed that they had a right to withdraw at any point during the study. One of the researchers double coded the transcripts to ensure the assignment of codes and themes was consistent across both researchers. In this study data quality checks or audits and peer review of coding were undertaken to verify accuracy of data analysis. Confirmability was ensured by both the first author and the research assistant checking and rechecking the emergent themes, the use of thick descriptive data to support emerging themes and comparing data with previous research findings.

### 2.6. Ethical Considerations

The study protocol was approved by the University of KwaZulu-Natal Biomedical Research Ethics committee (Approval Number: BE 675/17, 9 April 2017) and the KwaZulu-Natal Department of Health Ethics committee (Approval Number: HRKM Ref: 031/18, 14 February 2018). Study participants were provided with the information sheet explaining the objectives of the study. All participants signed informed consent forms prior to participation.

### 2.7. Findings

We report on the interviews held with twelve primary health care facility managers across the three NHI pilot districts in KwaZulu-Natal. Major thematic areas that emerged are role of operational manager, support and supervision, supplies, meeting place, lack of acceptance of CHWs/CCGs, access to transport, training and challenges.

### 2.8. Role of Operational Manager

The primary role of the operational manager (OM) is to support, supervise and to oversee the work of the WBOTs. She ensures that the teams are working efficiently, and checks that the planned schedules are being implemented.

“For the ward based teams my role is to ensure that teams have got all their resources so that they can go out and perform their duties. Then for the client; when the clients have complaints they must come and talk to me so that I can sort it out”[OM12]

It is the operational manager’ responsibility to ensure there are functional community- based forums, such as Operation Sukuma Sakhe and Phila Mntwana, the OM needs to ensure that the centres are equipped with the correct supplies. In addition, she oversees that the teams submit daily, weekly and monthly data on time. The OM is also in charge of the vehicles used by WBPHCOTs and needs to maintain them in good condition at all times. Accompanying the teams occasionally during field visits also falls within the role of the OM in the implementation of WBPHCOTs.

“Sometimes you have to go out with them when they are doing household registration just to see how they are doing, what difficulties do they encounter and help them solve those problems. The Phila Mntwana Centres we open them, like we ask for a room maybe from the community or school”[OM04]

### 2.9. Provision of Support and Supervision for Teams

The operational manager should hold regular meetings with the WBPHCOTs to discuss challenges, successes and ways of addressing gaps. The WBPHCOTs are seen as being part of the clinic team and not just the community-based team. During team meetings, targets and performance indicators are discussed and teams are able to see their importance in the outcomes of the clinic as a whole for all the programmes.

“We convene meetings with them. We also have discussions every morning during team briefings where they indicate if there is anything that they need. They also report their plans for the day ahead, where they will be working and so on. They have a roster. So I am also always aware of where they are on any given day”[OM02]

It is also during these meetings that the operational manager gets to learn about the experiences and needs of the team. In addition to the formal monthly meetings with the whole team, briefings are held before the team leaves the clinic for the community. Facility manager occasionally accompany the teams to support the activities in the community. However, managers shared that providing support and supervision to the outreach teams have some challenges given that they are not facility based.

“Sometimes I go with them to do door-to-door campaigns. If they are doing household registration, I go with them to see what problems they encounter. If they come back with problems, we have a meeting in our clinic every day, I mean every Friday, if possible. If there are no other meetings, I meet with the Team Leader, the team and Community Caregivers just to share the problems they encounter and then find a way forward [inaudible]”[OM04]

Inadequate supplies Inadequate and shortage of equipment and supplies impact negatively on service provision. The operational manager provides supplies to the WBPHCOTs. Sometimes there are challenges with procurement of supplies regarding turnaround times, which are long, and the requested orders return incorrect. This then poses a challenge to share the scarce resources with outreach teams, for example, gloves which CCGs need to carry out activities at a household level.

“our suppliers, they…… take long to come back. Most of the time they [CCGs] just think you don’t care” [OM07] “The problem we have is with pampers [nappies], because we no longer have them. They [DOH] said we must not give them pampers, they [households] must buy them with their pension/grant money. Yet before the CCGs would provide them because they know the people who are bedridden” [OM01]

“They do not have a loudhailer. However, when it comes to resources from the clinic like BP machines and everything, they have that, the weighing scales and everything”[OM05]

### 2.10. Meeting Place for the Teams

Meeting spaces are challenging in some clinics. The WBPHCOTs make use of the available space in the clinic to have their meetings, but they do not have a designated room. In clinics that have infrastructure challenges, these teams gather in a room to conduct their meetings while standing whereas in some clinics, WBPHCOTs are able to conduct their meetings in the facility boardroom if there is one.

“Hey, the infrastructure is a challenge… yeah, we do have but it is a limited space…yeah, we cannot fit all the chairs. We end up being on a standing position because of the space, just to allow all the people to gather in that room”[OM06]

### 2.11. Lack of Acceptance of Community Caregivers

Some community members do not accept CCGs and do not allow them access in their homes. Lack of trust poses significant challenges in the provision of necessary health care. In most instances, CCGs are community members who come from the same community that they are expected to provide services which further complicates the trust relationship.

“They report that sometimes they are not welcomed in the areas that they are visiting, sometimes they say that people are afraid to disclose their issues, especially to the community caregivers, because they [CCGs] are from the same communities”[OM10]

“Sometimes CCGs have a problem because some people do not want to let them in, they do not allow them inside their households”[OM01]

### 2.12. Access to Transport

Transport issues seemed to pose the most significant challenge for most of the operational managers. This hinders their ability to go out into the community to support teams during field visits.

“The transport that they are using is the thing; it is small so the roads here are gravel which makes it difficult to drive the small vehicle”[OM12]

“Yes, our main challenge I would say is the issue of transport. We currently have a double cab that is being used by the outreach team, the School Health and WBOT, which I feel is not conducive for the services”[OM11]

“Yeah they do have transport although it is not in a good condition. It is old now and they do have challenges when the transport is going for service, it takes about three [weeks] to a month”[OM09]

### 2.13. Training

Facility managers expressed that they were not trained nor orientated about WBPHCOTs and their roles were not clarified at the beginning of the programme.

“There was just no training but we were just called to the meeting to be told that we are going to speak to them [WBPHCOTs] because they operate under us…. they report to us and everything they do they just inform us of what is going to happen”[OM03]

In another district, the facility managers indicated that they [managers] did receive some training after the teams were already operating in the communities.

“When they [WBPHCOT] got into communities, they were battling a bit…. It [training] helped me a lot…. community entry and how to get accepted in the community…. how do I engage with other departments within the catchment area [Pause] to have a referral system to other departments so that people get comprehensive care not just health care…. Training also strengthened the importance of integrating with other structures in communities like Operation Sukuma Sakhe”[OM03]

### 2.14. Challenges

#### Workload

Facility managers felt that their high workloads make it difficult for them to be able to deal with all activities related to the facility and the WBPHCOTs.

“So I have a lot of work to do for the facility and then this thing the family health team; so that adds to the load. There is so much and at times I am unable to supervise them”[OM12]

“…Sometimes… uhm. I do feel that I am overworked because I have to look [after] the whole institution”[OM03]

### 2.15. Resource Constraints

Lack of funding and staff shortages were amongst the challenges faced by facility managers. This impacts negatively in their supervisory function.

“With the department it is difficult because they say there are no funds, there are no posts, so hmm…work with what you have [laughs]”[OM03]

### 2.16. Successes

Operational managers however shared various examples of some success stories indicating the work of the WBPHCOTs and their impact on the clinic performance on various indicators. These are summarised as follows;

The WBPHCOTs have been successful in tracing and reintroducing many defaulters, especially around HIV, into the health care system.

They have been vital in the distribution of condoms at the community level.

The WBPHCOTs have a significant presence in the War Rooms.

Through the WBPHCOTs the clinics are able to work hand in hand with the schools and by doing this, the road to health cards and immunisations of children are up to date.

The WBPHCOTs are a key component to the community-based campaigns.

The School Health Teams work together with the WBPHCOTs and this has had a positive impact in the community.

The WBPHCOTs have formed positive relationships with the community and the community have become aware of the important role that the WBPHCOTs play in the community.

Communities are happy to provide space within the community to allow for WBPHCOTs to render services.

WBPHCOTs are rendering services in the community which are in line with the services rendered at the primary health care level. This positively impacts on the facility headcount as well as the facility targets.

Through conducting regular home visits, WBPHCOTs are able to uncover sensitive issues needing urgent attention. In this way they are able to link communities with not only the primary health care facility, but also to other departments including South African Social Security Agency (SASSA), Department of Social Development, Department of Home Affairs and South African Police Services.

The community-based activities performed by the WBPHCOTs have an impact on the clinic statistics as well as on the overall performance indicators of the clinic. The WBPHCOTs contribute to the screening and tracing of patients as well as to the overall continuity of care of the patients.

## 3. Discussion

“The Municipal Ward-based Primary Health Care Outreach Teams (WBPHCOTs) form a pivotal part of South Africa’s PHC re-engineering strategy. The outreach teams consist of Community Health Workers (CHWs) led by a nurse and are linked to a PHC facility. CHWs assess the health status of individuals in the households. They provide health promotion education, identify those in need of preventive, curative or rehabilitative services, and health related counselling, and refer those in need of services to the relevant PHC facility”(Department of Health. National health insurance for South Africa: towards universal health coverage, 2015: 29–30) [8]

Studies have shown that community health workers can contribute significantly in improving lives of populations [2] Operational managers also referred to as facility managers in charge of primary health care facilities oversee activities of ward based outreach teams linked to their facilities. Amongst other functions, facility managers provide support and supervision to WBPHCOTs. However, this study has revealed that there are challenges with support and supervision of these teams since they are not facility based. Facility managers are overwhelmed with clinical and management responsibilities exacerbated by the burden of disease and human resource constraints [17]. This study showed that operational managers try to supervise the WBPHCOTs within the constraints they face. For example, operational managers generally hold monthly meetings and team briefings each morning before the WBPHCOTs leave the clinic to reach out to communities. However, operational managers confirmed that they seldom do field visits; this lack is felt especially when there are critical health related issues in the households. These findings are in agreement with the results of the study done in the Eastern Cape province that supervisors would meet weekly with community health workers and conduct field visits once a month. However, the reported perspective of some CHWs is that contact with supervisors occurs less often than reported here. In one study, CHWs observed that meetings with supervisors occurred once a month or less [18].

In this study it emerged that the supervision of WBPHCOTs at household level is difficult. Facility managers acknowledged that they have a supervisory role at household level, however, they cannot cope with facility supervision, clinical duties and community visits, and accompany CHWs on household visits only occasionally. However, facility managers’ roles in support and supervision of CHWs included training, household visits where there were complicated problems, direct observations and provision of supplies to CHWs to execute tasks at household level. A previous study has reported that supervisors spent more than two months without carrying out supervisory activities in the field. This was attributed to lack of resources, poor coordination within district health services and other logistical issues [19]. Another study has highlighted that the role of supervisors is to provide ongoing training of CHWs, mentoring and support, provide supplies, review their work and provide support [20]. This concurs with our study findings where participants indicated that they provide material supplies for WBPHCOTs and ensure that equipment and transport for the outreach teams are in good working order. However, transport for the teams was found in this study to be a major, ongoing problem. In availability of transport has a negative impact on field teams’ supervision by the facility managers. It also limits the frequency with which the teams go out to the field.

This study also documents challenges faced by WBPHCOTs, which amongst others include non-acceptance of CHWs by some households intensified by the fact that CHWs come from the same communities. Lack of training or orientation on the WBPHCOTs programme came out strongly from the facility managers. This is in agreement with a study done in Mozambique which confirmed lack of formal training of supervisors. In that study, as in this one, supervisors had to learn on the job [19]. Regardless of challenges encountered by the outreach teams, in which CHWs are frontliners, this study documented successes of these teams, which impact positively on the clinic outputs. The teams have resulted in a more seamless integration of health care between household and community level, which may be decreasing the patient load at clinics and improving overall health status (although this is not yet documented). A number of success stories learnt from this study include community-based activities which have an impact on the clinic statistics and on the overall performance indicators of the clinics. Identifying health problems at a household level, screening and tracing of patients, referrals, forming collaborations with other departments and working together with school health teams are some of the successes of these teams. Literature confirms that fixed clinic facilities form the foundation of the PHC system in South Africa [5]. Facility managers have strong bio-medical and clinical training and mainly provide curative services. However, these clinical services are not sufficient to address the social determinants of health. A wide range of skills, competencies and attributes are required to provide community and home based interventions, care and support [5]. Therefore, CHWs form an important and much needed cadre of staff in the health system in South Africa.

## 4. Limitations

The study only reports on the perceptions of facility managers regarding successes and challenges of ward based primary health care outreach teams. There is a need to document voices of outreach team leaders and community health workers on support and supervision they receive from supervisors.

## 5. Conclusions

There is a need to strengthen support and supervision of ward based primary health care outreach teams. This research revealed some inherent challenges for both the supervisors and the teams. Ward based primary health care outreach teams have a positive impact in preventive and promotive health in rural communities. Furthermore, these teams have also made impact in improving facility indicators despite the challenges inherent in their jobs. The study adds to the body of knowledge on ward based primary health care outreach teams and documents important success stories in communities and in primary health care facilities brought about by outreach teams. The study further highlights some challenges encountered by outreach teams, which are important to flag so that interventions can be put in place as remedies to convert challenges into successes.

## Figures and Tables

**Table 1 healthcare-09-01718-t001:** Profile of participants.

Participants	Designation	Years of Experience at PHC Facility
OM1	Facility manager	13 years
OM2	Facility manager	14 years
OM3	Facility manager	*
OM4	Facility manager	9 years
OM5	Facility manager	1 year
OM6	Facility manager	2 years
OM7	Facility manager	10 years
OM8	Facility manager	15 years
OM9	Facility manager	4 years
OM10	Facility manager	*
OM11	Facility manager	5 years
OM12	Facility manager	10 years

* Response not recorded.

## Data Availability

This is a qualitative study and the participants did not consent to having their full transcripts made publicly available. However, data on this study may be made available upon reasonable request to the relevant stakeholders and the authors: mhlongoem@ukzn.ac.za.

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
