# Peer review of "Facility Managers’ Perceptions of Support and Supervision of Ward Based Primary Health Care Outreach Teams in National Health Insurance Pilot Districts in KwaZulu-Natal, South Africa. A Qualitative Study"

_healthcare, 2021, doi:10.3390/healthcare9121718_

Round 1
Reviewer 1 Report
Authors are well known with the health sector in South Africa. They published two articles about the WBPHCOT’s in South Africa: a scoping review protocol and the roles, responsibilities, and perceptions of community health workers. It’s my impression that the actual article is part of a larger study on the functioning of the community health care.
In the introduction an extensive description of the need and role of outreach teams is described. But the aim of the study is not yet clear. In the introduction authors cited ‘supportive supervision’ the golden standard but they even didn’t give a definition. Subsequently they called the purpose of the study ‘managers perception of supervising’. And in the paragraph ‘aim of the study’ they called it ‘support and supervision’. My suggestion to the author’s is to shorten the text about the need of outreach teams (that’s context but not the subject of this paper) and to define more clearly what is the aim/purpose of the study.
In the paragraph on ‘key concepts’ you have to define support and supervision/supportive supervision? You can eliminate the actually described key concepts because they aren’t essential for this paper. In some way the ‘golden standard’ of the key concepts will describe the role, tasks and functioning of the facility managers in ideal circumstances. In the chapter results this ‘golden standard’ can be compared to the perceptions of the facility managers how in practice the system is functioning.
In the title of the manuscript the main subject ‘perceptions of support and supervision’ must be mentioned.
Some minor remarks:
Paragraph on data analysis. It’s seeming to me that in the first sentence a word is lacking: …… is necessary to establish (what?) of the study……
Paragraph on trustworthiness: It seems to me that the text came from another article. In this study only facility managers were questioned and not the community.
Paragraph on the role of operational manager. What is the role: support, supervise or (???) to oversee. Isn’t it support, supervise and to oversee…….
In the text it isn’t always clear if it is a citation. Citation must be clearly marked.
Paragraph discussion, last sentence: ‘Therefore, CHW’s form an important etc.’ This conclusion can’t be drawn from the actual results of this paper. The paper wasn’t about the functioning of CHW’s.
I suggest limiting the conclusion to the purpose of the study on support and supervision.
Author Response
Dear Reviewer
Please find the attached file with responses to reviewer's comments
Regards
Mbali Mhlongo

Reviewer 2 Report
Thank you for the opportunity to review this article. The linkages between community health work(ers) and clinical settings are an important area of research.
Abstract:
The two concluding sentences are vague. Refine these to provide more concrete implications.
Introduction:
The introduction should be split into a few different paragraphs.
When you refer to the beginning of democracy remind readers when this was.
In the first paragraph of the introduction you introduce some of the acronyms multiple times. Just once is enough.
A timeline with several of the events of the first paragraph might be helpful. Be sure to clearly define when WBPHCOTs began implementation.
Is rPHC a typo?
“Supportive supervision is con-sidered the gold standard practice in CHW supervision” This is an odd sentence. What would the alternative be, non-supportive? The focus on supervision seems a bit underdeveloped and perhaps unnecessary. Readers may be more interested in the challenges of implementing this new initiative connecting communities to care with CHWs that are part of clinics.
In the last sentence, are you referring specifically to supervision of CHWs?
Some more background on CHWs and their roles in connecting to the community (and not just in S. Africa) might be helpful.
In the introduction you should highlight what this work does that is new or different than others.
Reframe the definition of key concepts section to be included in the introduction. Some of it seems redundant.
More history on the pilot project is needed. Why were these sites selected for the pilot? Where else has the program been piloted?
Methods:
It would be helpful to see the interview guides as an appendix.
Why did you select your sites? Are there other sites where pilot projects are also underway? If so, why did you focus in these districts?
Did you interview all facility managers with WBPHCOTs (for over 3 months) or only some of them? If only some, how were they selected?
It will be helpful to understand how your sample fits within the larger scope of places where this program was being implemented at the time.
Did you use any qualitative software for the analysis?
Explain your double coding process. This may be a different definition than I am used to for double coding.
Findings:
Consider a different way of listing themes. For example, “supplies” doesn’t sound like a theme, nor do some of the others. What is it about supplies that makes it important? Something like “Insufficient access to supplies presents a barrier to program implementation” (I’m not suggesting using this) provides a better name for a theme.
Make sure it is clear when something is a quote and when it is just part of a paragraph.
The role of operational manager doesn’t seem like an important theme either. It’s simply background information that could be provided earlier without quotes.
The way the themes and quotes are presented make this article very specific to the implementation of this program’s operations. Some of the quotes (we have meetings on certain days) are not particularly helpful for a broad audience.
In the findings and results section, interspace the quotes with commentary explaining each quote and how it relates to the theme.
Successes should be written out (with exemplary quotes) and not summarized in bullet points.
Discussion:
The first paragraph should be in the introduction.
You reference another study in Eastern Cape. This should also be discussed in the introduction. What does this study tell us that that one did not? How is it different?
The discussion should be revised once you have reviewed your main findings.
Some of the articles that you cite in the discussion seem like they are from such similar work that they should be identified and described in the introduction, along with how this work adds to what was already found.
Limitations:
You are also only in a very specific region of the country.
Conclusion:
“Strengthening support” is not a compelling conclusion.
This article could use some grammatical review. More importantly, revise the findings and put this work in the context of the global literature on community outreach initiatives that are tied to clinics/wards.
Author Response
Dear Reviewer
Please find attached the file with responses to reviewer's comments
Regards
Mbali

Round 2
Reviewer 1 Report
The authors managed adequately the major comments from reviewer 1 and 2. However.
The article is rather long and didn't give a thorough analysis of supervision in all his aspects:
-Details on how the elements of support and supervision were executed: training, field visits, direct observation, delivery of materials, etc.
-What are the major constraints: time? money? capacity? transport? buildings? etc.
-Details on the opinion of the community workers on support and supervision; this element was cited in the definition of supportive supervision.
One other remark: the conclusions in the abstract are not harmonized with the conclusions in the main text.
Author Response
1. Details on how the elements of support and supervision were executed: training, field visits, direct observation, delivery of materials, etc.
Response: A paragraph was added in the discussion to address this
2. What are the major constraints: time? money? capacity? transport? buildings? etc.
Response: These have been highlighted in the results section of the paper
-Details on the opinion of the community workers on support and supervision; this element was cited in the definition of supportive supervision.
Response: This paper seeks to highlight the facility managers' perceptions of support and supervision of WBPHCOTs and not merely seeking the opinions of CHWs.
One other remark: the conclusions in the abstract are not harmonized with the conclusions in the main text.
Response: Conclusion paragraph revised
Reviewer 2 Report
The conclusions from this study remain vague and would be difficult to apply to other settings outside of this context. It would be beneficial to have more lessons learned with practical implications/suggestions for improvement so that this work would be seen as more relevant to a wider audience.
The minor revisions made by the authors were insufficient to overcome some of the challenges described after the first review.
Author Response
The conclusions from this study remain vague and would be difficult to apply to other settings outside of this context. It would be beneficial to have more lessons learned with practical implications/suggestions for improvement so that this work would be seen as more relevant to a wider audience.
Response: Discussion and conclusion improved
The minor revisions made by the authors were insufficient to overcome some of the challenges described after the first review.
Response: Reviewer's comments attended to